# An initial 'snapshot' of sensory information biases the likelihood and speed of subsequent changes of mind

William Turner ®*, Daniel Feuerriegel ®, Robert Hester, Stefan Bode®

Melbourne School of Psychological Sciences, The University of Melbourne, Melbourne, Australia

* wturner@student.unimelb.edu.au

## Abstract

We often need to rapidly change our mind about perceptual decisions in order to account for new information and correct mistakes. One fundamental, unresolved question is whether information processed prior to a decision being made ('pre-decisional information') has any influence on the likelihood and speed with which that decision is reversed. We investigated this using a luminance discrimination task in which participants indicated which of two flickering greyscale squares was brightest. Following an initial decision, the stimuli briefly remained on screen, and participants could change their response. Using psychophysical reverse correlation, we examined how moment-to-moment fluctuations in stimulus luminance affected participants' decisions. This revealed that the strength of even the very earliest (pre-decisional) evidence was associated with the likelihood and speed of later changes of mind. To account for this effect, we propose an extended diffusion model in which an initial 'snapshot' of sensory information biases ongoing evidence accumulation.

**Data Availability Statement:** All data and analysis/ modelling code for this paper are available at https://osf.io/a6u4n/.

## Author summary

To avoid harm in an ever-changing world we need to be able to rapidly change our minds about our decisions. For example, imagine being unable to overrule a decision to run across a street when you realise a speeding car is approaching. In this study, we examined the information processing dynamics which underlie perceptual judgements and changes of mind. By reverse correlating participants decisions with the moment-to-moment sensory evidence they received, we show that the very earliest information, processed prior to an initial decision being made, can have a lasting influence over the speed and likelihood of subsequent changes of mind. To account for this, we develop a model of perceptual decisions in which initial sensory evidence exerts a lasting bias over later evidence processing. When fit to participants' behavioural responses alone, this model predicted their observed information usage patterns. This suggests that an initial 'snapshot' of sensory information may influence the ongoing dynamics of the perceptual decision process, thus influencing the speed and likelihood of decision reversals.

**Funding:** This work was supported by an Australian Research Council (ARC) Discovery Project Grant [DP160103353] to S.B. and R.H. (https://www.arc.gov.au/grants/discovery-program/discovery-projects) and an Australian Government Research Training Program (RTP) Scholarship to W.T (https://www.education.gov.au/research-training-program). The funders had no role in study design, data collection and analysis, decision to publish, or preparation of the manuscript.

**Competing interests:** The authors have declared that no competing interests exist.

## Introduction

The ability to rapidly revise decisions in the face of new information is critical for avoiding harm in an ever-changing world. For example, imagine being unable to overrule a decision to run across a street when you realise that a speeding car is approaching. In situations such as this, even small delays in the time it takes to change your mind can have serious consequences. Given this, it is important to understand the cognitive processes which underlie rapid decision reversals, and the factors which influence the likelihood and speed with which they occur.

There is extensive evidence that perceptual decisions are made via the noisy accumulation of sensory information over time [1,2]. Once a certain amount of evidence has been accumulated in favor of one choice over another, an initial decision is made. However, following this the decision process does not immediately halt. Instead, sensory evidence continues to be accumulated, and, if enough subsequent evidence is accumulated in favor of the initially-unchosen response, a change of mind occurs [3].

Relative to the point at which an initial decision is made, two broad time periods can be defined: 1) a pre-decisional period, in which sensory evidence is being accumulated but a decision is yet to be reached, and 2) a post-decisional period in which a decision has been made but the evidence accumulation process is continuing to unfold. When considering these two time periods, one fundamental question which arises is whether information processed in the pre-decisional period ('pre-decisional evidence') has any influence on the likelihood and speed with which a change of mind subsequently occurs. Intuitively, it is appealing to think that pre-decisional evidence will affect subsequent change-of-mind behaviour. For example, if a decision is made on the basis of strong sensory evidence then it is reasonable to assume that this decision will be less likely to be overruled, or, if it is overruled, it will take longer for this to occur. However, it is also possible that change-of-mind decisions are be based solely on information processed after an initial decision has been made. Indeed, the most prominent model of perceptual changes of mind, the extended diffusion decision model [3], makes just this prediction. In particular, this model proposes that (controlling for decision accuracy) the decision process is in exactly the same state at the time of the initial decision, meaning that only post-decisional evidence influences change-of-mind speed and likelihood. It is therefore important to test between these two possibilities, to better understand how we rapidly change our minds about perceptual decisions.

Previous studies that investigated how people make discrimination judgements about dynamic stimuli (e.g., random dot motion stimuli or flickering dot arrays) have shown that, in trials where people change their mind, the sensory evidence provided by the stimulus initially favors one decision. However, just prior to the behavioral response being enacted, the evidence switches to favoring the alternative option, driving the change of mind [3–5]. It has been argued that, when controlling for initial response accuracy, the likelihood of a change of mind occurring depends on how strongly the initial decision was supported by the pre-decisional evidence [4]. However, in the study which purported to show this, sensory and motoric delays were not accounted for. Because it takes time for the brain to process incoming sensory information and output a motor response, decisions are actually made on the basis of information which was presented some hundreds of milliseconds earlier in time, and behavioural responses lag behind decisions by ~80 ms [3]. Since these delays were not accounted for, it is unclear whether the reported association was driven by truly pre-decisional evidence, or rather evidence which was presented prior to a behavioural response being made, but after the point at which it could have influenced the initial decision. Beyond the question of whether pre-decisional information affects change of mind likelihood, the associated question of whether this information influences more detailed response characteristics, such as change-of-mind speed,

has also not been explored. Understanding the influence of pre-decisional information on the speed with which decisions are overruled is important because the timing of decisions often offers a rich source of information about the underlying decision process [6].

In previous studies of change-of-mind behaviour, random dot motion tasks have most often been employed [3,5,7,8]. To examine information usage in these tasks motion energy filtering [9] is required, leading to a smearing of information across time that prohibits the examination of early information usage due to a 50–150 ms lag in filter build up. To sidestep these issues, we employed a dynamic luminance discrimination task in which it is possible to examine information usage independently on a frame-by-frame basis, without filtering. To allow for fine-grained estimation of the participants' sensory information usage patterns (their 'psychophysical kernels'), we employed a small-N design in which each participant (n = 4) completed a very large number of trials [10]. Participants indicated which of two flickering grey squares was the brightest by pressing one of two response keys. After an initial judgment, the stimuli remained on screen for a brief period (1.5 seconds), and the participants were free to change their response. Critically, with each screen refresh (i.e. every 13.33 ms) a random luminance value was added to the mean luminance value of each square. Using psychophysical reverse correlation [11,12] we then retrospectively examined the impact that this residual evidence had on participants' decisions, on a frame-by-frame basis. If the luminance fluctuations at each frame are averaged across all trials they will cancel to zero, due to the fact that they are randomly distributed with a mean of zero. However, if the fluctuations systematically affect participants' decisions, then averaging across trials with shared decision outcomes (e.g., averaging over correct responses) will reveal when and how the fluctuations influenced participants' decisions across the pre-decisional and post-decisional time periods [13]. Below, we first describe the psychophysical kernels that this reverse correlation analysis revealed. Then, we account for these with a variant of the extended diffusion decision model.

## Results

Participants' changed their mind on 23.91% of trials (18.53% corrected errors and 5.38% spoilt responses) with an average initial response accuracy of 66.95% (80.10% of final responses were correct). To investigate whether pre-decisional evidence affected change of mind likelihood we sorted trials into four possible types: correct and error responses that were not followed by a change of mind, and correct and error responses that were followed by a change of mind (termed 'spoilt correct' responses and 'corrected error' responses, respectively). We then time-locked the residual evidence relative to: 1) stimulus presentation, 2) the initial response, and 3) the change of mind response, and averaged across trials within each response type (Fig 1A–1C). This revealed clear patterns in the luminance residuals for change of mind trials. Evidence in early time windows favoured the initial response, whilst evidence in later time windows favoured the final response, replicating previous findings [3–5]. This can be most clearly observed in Fig 1C, where the psychophysical kernels for 'corrected error' and 'spoilt correct' trials slowly shift from favouring one choice outcome to the other over the course of ~1 second. Interestingly, the kernels both remain significantly different from zero even after the change of mind was enacted. This is likely due to the exclusion of double-change-of-mind trials from our analyses. That is, additional evidence in favour of the revised response may have been needed in order to avoid another secondary change-of-mind from occurring, and trials with secondary changes of mind were excluded from these plots.

Strikingly, the reverse correlation analyses also revealed that the strength of evidence favoring the initial response, for even the very first frame of evidence presented, differed depending on whether or not participants ultimately changed their mind (Fig 1A). To formally examine

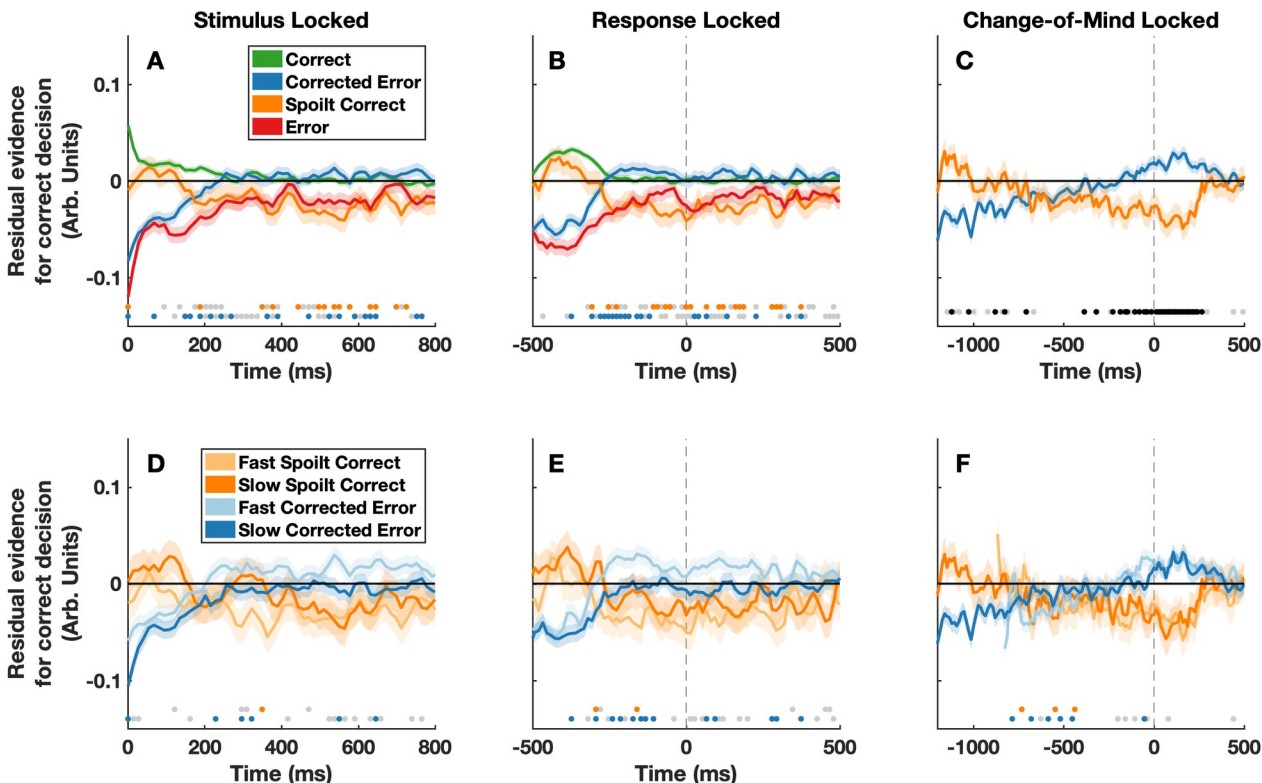

**Fig 1. Psychophysical reverse correlation results.** Panels A) and D) show the residual stimulus fluctuations time-locked to stimulus onset. Panel A) shows correct responses (green), error responses (red), corrected errors (blue) and spoilt correct responses (orange). Panel D) shows fast (light blue) and slow (dark blue) corrected errors, and fast (light orange) and slow (dark orange) spoilt responses. Panels B and E show the same data, time-locked to the initial response (the time of the initial response is indicated by the grey dashed lines). Panels C and F show the data from trials containing a change of mind, time-locked to the change-of-mind response (the time of the change-of-mind response is indicated by the grey dashed lines). In all panels positive values on the y-axis indicate more evidence supporting the correct response. For display purposes a moving average smoothing function with a span of 3 frames was applied and a median split for change-of-mind response time was used in panels D-F. However, statistical analyses were conducted on the unsmoothed, continuous data. For illustrative purposes, t-tests (alpha level of .01) were also conducted to compare levels of residual evidence between specific trial types, at each frame. Residual evidence values were pooled across participants for these tests. Orange dots denote timepoints at which there was a significant difference between correct and spoilt correct trials (Panels A-B), or where there was a significant difference between fast and slow spoilt trials (Panels D-F). Blue dots denote timepoints at which there was a significant difference between error and corrected error trials (Panels A-B), or where there was a significant difference between fast and slow corrected error trials (Panels D-F). Finally, black dots (Panel C) denote significant differences between corrected errors and spoilt correct responses, and grey dots (all panels) denote time points at which there was a significant difference for an alpha level of .05.

this effect, we fit a generalized linear mixed effects model to predict the probability of a change of mind occurring. This revealed a significant interaction between initial decision accuracy and the strength of the first frame of evidence on the probability of a change of mind occurring (likelihood ratio test, $\chi^2 (1) = 42.08$, $p < 8.75 \times 10^{-11}$, see Table A and Fig A in S1 Text). This indicates that when the very first frame of evidence strongly supported participants' initial decisions, then changes of mind were less likely to occur. Note, here we have restricted our analyses to just the initial frame of evidence, as a pre-decisional evidence source. However associations between stimulus evidence strength and decision behaviour were by no means restricted to just the first frame (see below for additional analyses regarding the effects of sub-sequent evidence). For illustrative purposes, we also plot the results of frame-by-frame t-tests to compare differences in residual evidence throughout the trial (see Fig 1).

Having found that the strength of the very first frame of evidence was associated with change-of-mind likelihood, we then investigated whether the strength of this information was

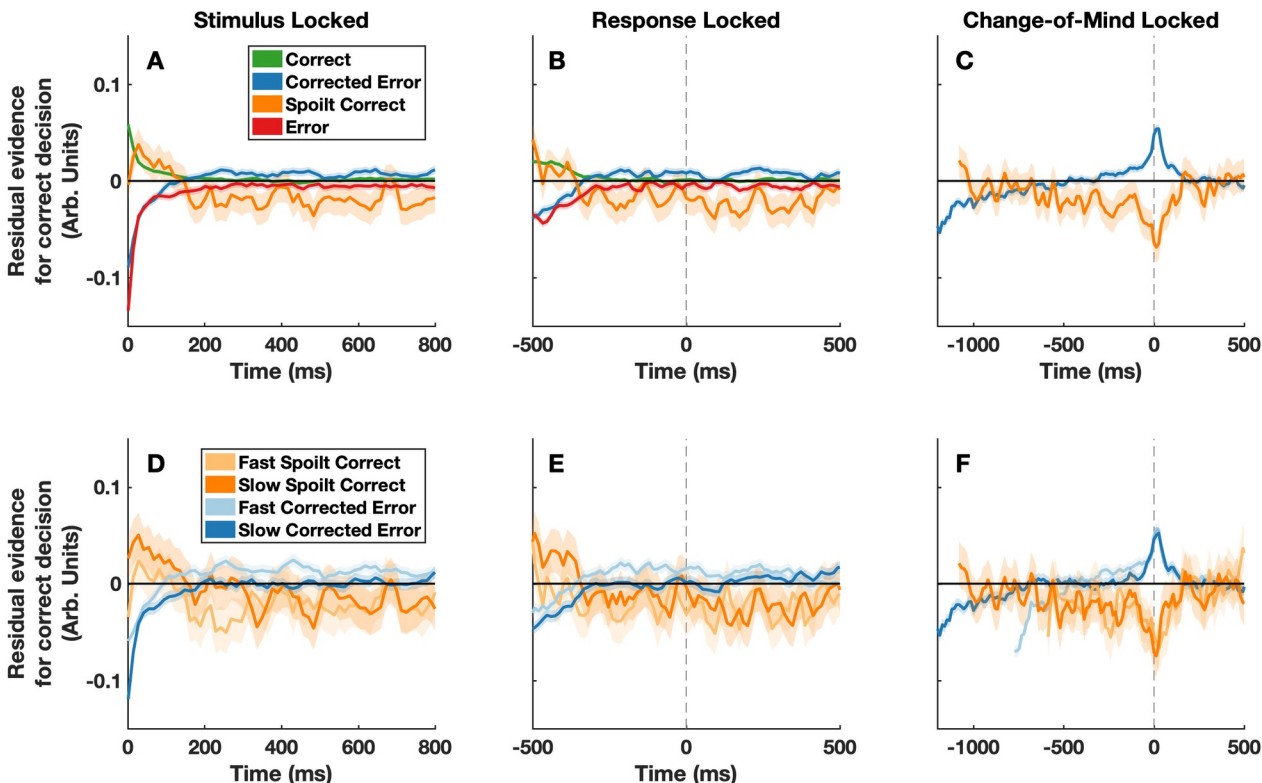

**Fig 2. Model-derived psychophysical kernels.** Panels A-F show the model predicted psychophysical kernels. To create these kernels we simulated decision variable trajectories in 100,000 experimental trials. We then took the within-trial noise in the model–used to simulate the moment-to-moment fluctuations in stimulus luminance–and reverse correlated this with the predicted response outcome on each trial. All plotting conventions are the same as in Fig 1.

associated with the speed with which changes of mind occurred. Plotting the residual stimulus fluctuations for fast and slow corrected errors and fast and slow spoilt responses revealed that changes of mind occurred more slowly after the initial response when the first frame of evidence strongly supported the initial decision (Fig 1D–1F). Note, at around -500 ms in Fig 1F it may appear as if there is actually stronger evidence in favour of the initial response for fast change-of-mind trials, as compared to slow change of mind trials. However, this is simply due to a shifting of the relative timing of kernels, given the underlying differences in change-of-mind response time (see Fig 2F for a model-based recreation of this). To formally test the effect of initial evidence strength, we fit a linear mixed effects model to predict the speed with which changes of mind occurred. This revealed a significant interaction between initial decision accuracy and evidence strength on change-of-mind speed (likelihood ratio test, $\chi^2$ (1) = 4.27, $p$ = .039, see S1 Text). This indicates that changes of mind were slower when the very first frame of evidence strongly supported participants' initial decisions.

Overall, these analyses demonstrate that the likelihood and speed with which participants changed their mind was already associated with the strength of even the very first frame of sensory evidence they saw (i.e. a pre-decisional source of information). These effects were consistent on the individual level and were symmetric across the two stimuli (reported in S2 Text).

In addition to the main analyses reported above, we also examined the overall effect of pre-response evidence on the speed and likelihood of changes of mind. In particular, we fit a generalized linear mixed effects model to predict the probability of a change of mind occurring from the average pre-response evidence. Note, when calculating pre-response residual

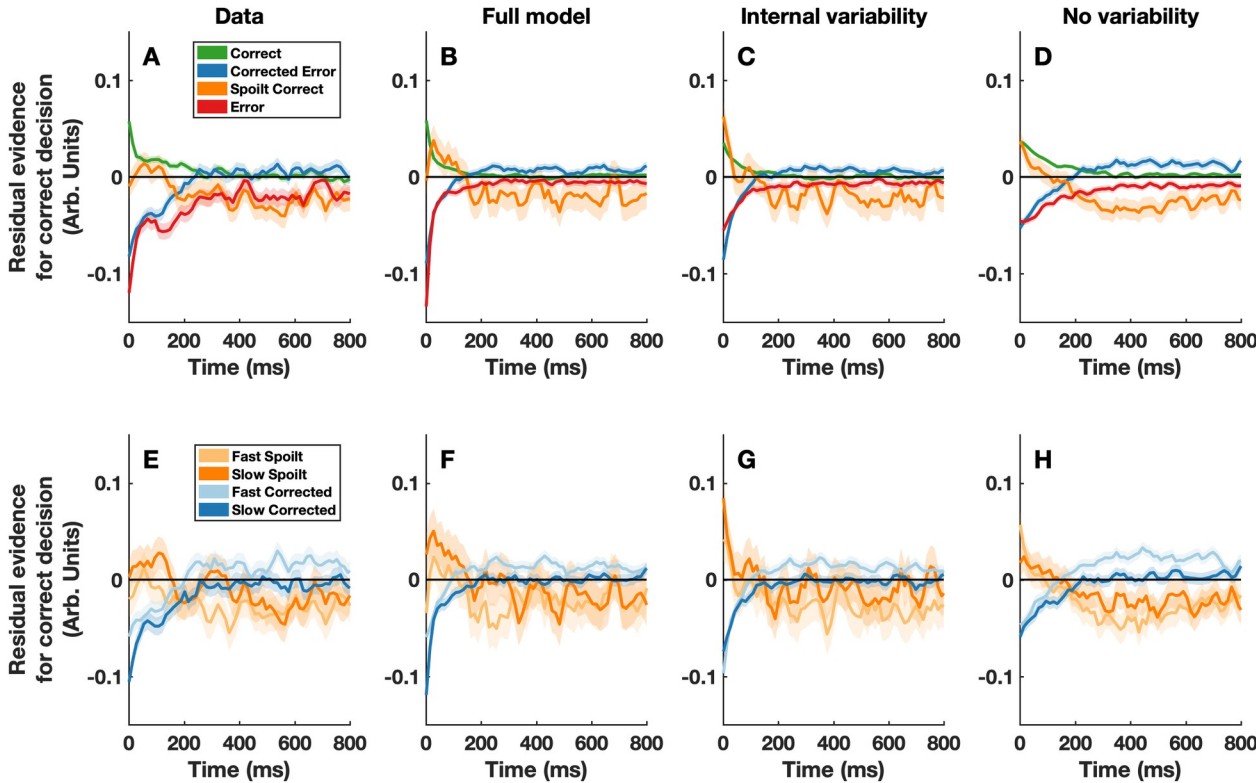

**Fig 3. Actual and predicted stimulus-locked psychophysical kernels.** Panels A and E depict the psychophysical kernels calculated from the data. To create panels B-C and F-G, we simulated 100,000 trials from three variants of the extended DDM model. The first variant (panels B and F) contained both external and internal sources of across-trial drift rate variability (same as in Fig 2). For the second variant (panels C and G) across-trial drift rate variability had exactly the same distribution, but was de-coupled from the stimulus (i.e. purely internal). Finally, for the third model variant (panels D and H) we simulated a version of the extended DDM in which drift rate was constant across trials. After simulating each model, we took the stimulus driven within-trial noise–used to simulate the moment-to-moment fluctuations in stimulus luminance–and reverse correlated this with the response outcome on each trial. All plotting conventions are the same as in Fig 1. See Fig B in S3 Text for a plot of this data for the first frame only (i.e. at x = 0). We note the slight apparent differences between the kernels in the first frame of Fig 3D and 3H, the largest being between fast and slow spoilt correct responses. These are not indicative of a true effect and arise due to the relative rarity of these trials (e.g., the rarity of spoilt correct responses). Across repeated simulations of the model, these differences are not consistent and will average to zero. Importantly, these differences highlight two points which it is important to be mindful of when conducting and interpreting psychophysical reverse correlation analyses. Firstly, they demonstrate the importance of aiming for maximum practicable statistical power, given the inherent noisiness of this technique. Secondly, they highlight the importance of not placing strong weight on single effects, but rather interpreting collective patterns of differences which are consistent across individuals.

evidence, we excluded the initial frame. This allowed us to investigate the impact that subsequent sensory evidence had on change-of-mind behaviour, to get a general sense of the relative importance of the 'primacy effect' we report above. This analysis revealed a significant interaction between initial decision accuracy and pre-response evidence strength (likelihood ratio test, $\chi^2(1) = 192.36$, $p < 2.20 \times 10^{-16}$; see Fig A in S4 Text). This indicates that when pre-response evidence (excluding the initial frame) strongly supported participants' initial decisions, then changes of mind were less likely to occur. To formally test the effect of pre-response evidence on change of mind latency, we fit a linear mixed effects model to predict the speed with which changes of mind occurred. This revealed a significant interaction between initial decision accuracy and subsequent pre-response evidence strength (likelihood ratio test, $\chi^2(1) = 82.36$, $p = 2.20 \times 10^{-16}$; see Fig A in S4 Text).

Importantly, these analyses show that the initial frame of evidence is neither the sole, nor necessarily the best, predictor of change of mind behaviour. Indeed, under the current

experimental conditions, mean evidence between 200–400 ms after stimulus onset was a better predictor of change of mind likelihood than mean evidence between 0-200ms (non-nested GLMM comparison: AIC 14961 vs. AIC 15039; Marginal $R^2$/Conditional $R^2$ 0.370/0.372 vs. 0.364/0.367, see Supplement 6 for model parameter estimates). Considering the patterns revealed by the psychophysical reverse correlation analysis, as well as the results of these additional analyses, it is clear that changes-of-mind arise due to stereotyped changes in the sensory evidence across many timepoints. Clearly, most frames of evidence are able to contribute to these decisions in some way, with late arriving evidence potentially have a greater impact than early evidence. However, as we demonstrate below, the fact that the very earliest frame of evidence can even influence change-of-mind behaviour at all, regardless of the magnitude of this effect, is theoretically interesting and reveals important details about the structure of the processes underlying change of mind decisions.

## Computational modelling

Having found that the strength of even the very first frame of evidence was associated with the likelihood and speed of subsequent changes of mind, we then sought to account for this effect within a computational framework to provide a mechanistic account of our findings. To this end, we employed the most prominent model of change-of-mind behaviour, the extended diffusion decision model (extended DDM; [3]). In this model, decisions are made by noisily accumulating the relative evidence between the two choice options (i.e. the difference in luminance between the two squares) to a threshold level. Once an initial decision has been made, sensory evidence continues to be accumulated. If enough evidence is accumulated against the initial decision, such that a separate change-of-mind threshold is crossed, then a change of mind occurs.

## The original extended DDM predicts an insensitivity to pre-decisional evidence

In the original version of the extended DDM [3], the drift rate (i.e. the rate at which sensory evidence is accumulated over time) is assumed to be constant across trials of matched stimulus difficulty. As such, the only source of variability within the decision process is within-trial noise (i.e. moment-to-moment fluctuations in the decision process). Critically, this leads to the prediction that pre-decisional evidence will have no influence on change-of-mind behaviour, in direct contrast to the current findings (see Fig 3D and 3H). This is because, when controlling for initial decision accuracy, the decision process in the extended DDM is always in the same state at the time of the initial decision (i.e. there is always the same amount of accumulated evidence). As such, change-of-mind decisions depend entirely on the quality of post-decisional evidence. However, in the fields of cognitive and mathematical psychology it is typical to assume that the drift rate varies from trial to trial [14]. This enables the DDM to account for the relative timing of correct and error responses, and, as we will demonstrate below, under this assumption the extended DDM no longer predicts an insensitivity to pre-decisional information (see Fig 3).

## A novel variant of the extended DDM

To account for the current findings, we fit a variant of the extended DDM which included trial-to-trial variability in drift rate. In particular, we assumed that there are both external (stimulus-driven) and internal (endogenous) sources of across-trial drift rate variability. We assumed that the external variability component is a function of the residual evidence in the first frame of each trial, which linearly decreases over time (i.e. a decaying 'snapshot' of initial

evidence). This means that when initial sensory evidence strongly supports one decision over another, the drift rate is temporarily biased in favor of that decision. This is consistent with recent evidence suggesting that across-trial variability in drift rate is partly stimulus driven [14]. To model the effect of trial-wise differences in internal states (e.g., attention, motivation, and arousal), we included an internal drift-rate variability component which was combined in an additive fashion with the externally-driven component. Finally, we assumed that moment-to-moment variability in the decision process (i.e. within-trial variability) was driven by both moment-to-moment variability in the stimulus as well as endogenous variability. This allowed us to reverse correlate the simulated moment-to-moment variability in the stimulus with the model-predicted behaviour, to construct model-based psychophysical kernels, which could then be compared to the observed psychophysical kernels.

We fit our variant of the extended DDM to the response proportions and response time quantiles for initial and change-of-mind decisions simultaneously (see Fig A in S3 Text). Strikingly, after being fit to just the behavioural responses, the model was able to predict the observed patterns in the psychophysical kernels (e.g., weaker initial evidence in favour of the initial response on change of mind trials; Fig 2). Importantly, when trial-to-trial drift rate variability was de-coupled from the first frame of evidence, but was otherwise identically distributed, the model was no longer able to capture the observed patterns in the psychophysical kernels (Fig 3C and 3G). In this case, the predicted pattern for the first frame was in the opposite direction to the observed results (i.e. stronger initial evidence in favor of the initial choice on change of mind trials). Finally, when there was no across-trial variability in drift rate the model predicted that there would be no differences in the psychophysical kernels for the first frame of evidence (Fig 3D and 3H). By comparing these predictions, it is clear that stimulus-driven across-trial drift rate variability is the key feature within our variant of the extended DDM that allows it to capture the patterns in the psychophysical kernels. In simple terms, this indicates that the patterns we observed in the psychophysical kernels can be explained by an initial 'snapshot' of evidence exerting a slowly decaying bias on ongoing evidence accumulation.

## Discussion

We investigated whether information processed prior to a perceptual decision being made ('pre-decisional information') influences the likelihood and speed of subsequent changes of mind. Participants made comparative luminance judgements between two flickering grey squares and indicated their decision with a button press. Following an initial decision, the stimuli remained on screen for a brief period and participants could change their mind. Using psychophysical reverse correlation, we examined the effects of moment-to-moment random fluctuations in luminance on participants' decisions. We found that the likelihood and speed with which participants changed their mind was reliably associated with the strength of the very first frame of evidence they saw. To account for this observation, we developed a variant of the extended diffusion decision model (extended DDM), in which across-trial variability in drift rate is partially driven by the first frame of sensory evidence. Fitted to just the behavioural responses, this model was able to predict the observed patterns in the psychophysical kernels. Broadly put, this suggests that an 'initial snapshot' of sensory evidence exerts a slowly decaying bias on decision evidence accumulation, thus influencing later self-corrective behaviour.

The current study extends a nascent line of research examining the time-course of information processing underlying perceptual changes of mind [3–5]. One previous study has attempted to address the question of whether pre-decisional information affects change-of-mind behaviour [4]. However, because sensory and motoric delays were not accounted for, it was unclear whether the reported effect–a negative association between pre-decisional

evidence strength and change-of-mind likelihood–was driven by evidence which was truly pre-decisional. Other studies investigating perceptual changes of mind have typically employed random dot motion tasks, making it impossible to examine the frame-by-frame processing of early sensory information in these experiments due the 50–150 ms lag induced by motion energy filtering. By employing a luminance discrimination task, we were able to circumvent this issue and present clear evidence that change-of-mind behaviour is affected by even the very earliest (pre-decisional) information one receives.

Importantly, it may be the case that the 'primacy effect' we observed is only detectable in the context of low-level visual tasks which require minimal temporal integration. Similarly, the fact that the timing of stimulus onset was predictable in the present study, may have led participants to attend more to early information. If future work identified a differential use of information for change of mind, depending on the features of evidence integration, and/or the predictability of stimulus timing, this would in itself advance our understanding of decision dynamics. This would also help to clarify the degree to which change-of-mind behaviour is influenced by initial sensory evidence under different environmental conditions or decision scenarios–something which is of relevance when considering real-world decisions which rely on either rapid or more sustained integration of sensory information (e.g., gauging the color of a traffic light in clear conditions compared to judging the speed of approaching traffic in heavy rain). Nevertheless, given the dominant view that relatively general computational processes underlie simple perceptual decisions and changes of mind across different perceptual tasks [3,15,16], the effect we demonstrate also has general implications. In particular, it demonstrates that the pre- and post-decisional dynamics are not distinct, and do in fact interact, counter to dominant perspectives [3].

A novel variant of the extended DDM was able to recreate the patterns we observed in the psychophysical kernels, after being fit to just the behavioural response data [3]. In this model, trial-to-trial variability in drift rate is partly driven by a linearly decreasing function of the first frame of sensory evidence one receives (the fitted slope parameter causes this to decay over the course of ~1–1.5 seconds). This is consistent with the emerging view that a combination of both internal and external drift rate variability components is necessary to explain behaviour on perceptual decision tasks [14]. Comparing the predicted psychophysical kernels from our model to an otherwise identical model in which drift-rate variability is identically distributed, but de-coupled from the stimulus, it is clear that a coupling between the first frame of sensory evidence and drift-rate variability is the critical assumption for capturing the observed patterns in the psychophysical kernels.

We chose the extended DDM as a modelling framework because it is the most prominent model of change of mind behaviour, and because the parameters clearly link to different aspects of decision-making processes and are thus readily interpretable. However, there are other more complex–and arguably more biologically plausible–models which have been proposed to explain perceptual changes of mind [17,18]. Previous work has shown that under certain parameterisations of these models a 'primacy effect' occurs whereby early sensory information is weighted more heavily than later sensory information [13,19]. On face value this is similar to the bias exerted by the initial frame of evidence in the extended DDM variant in this study. Given this, it is possible that the initial evidence bias we have implemented in the current model, is in fact mimicking the primacy effect displayed by these more complex models. Future studies could examine the degree to which these models mimic one another, to uncover potential neural mechanisms through which initial sensory information might bias ongoing evidence accumulation.

One limitation of the current modelling approach is that, while our variant of the extended DDM was able to predict the weighting of the earliest frames of evidence (i.e. the experimental

effect of interest), it did not perfectly predict all observed patterns in the psychophysical kernels. For example, it predicted sharper peaks in the kernels around the time at which changes of mind occurred (compare Fig 1C and 1F to Fig 2C and 2F). This discrepancy may simply be due to the fact that the model was not directly fit to the psychophysical kernels. Alternatively, this may be a general limitation of the modelling framework or specific simplifying assumptions we adopted. For example, we assumed that the distributions of non-decision times for initial and change of mind responses were identical and were normally distributed. However, recent work has questioned whether alternative distributions of non-decision times might better explain behavioural data [20]. Ultimately, our modelling approach is constrained by the knowledge and data that are currently available to us and it is possible that future work may identify some complex alternative mixture of effects/assumptions within an evidence accumulation framework which offers a different explanation for the psychophysical kernel we observed. As such, we are not claiming to have necessarily discovered the 'one true model' for the current data, but instead, are simply aiming to provide a coherent account of the novel effects we have observed, with the goal of fostering further exploration in this area.

We chose to keep the means of the two stimuli fixed, and did not systematically vary their overall luminance. This was to maximise the number of useable trials in the psychophysical reverse correlation analyses. Because the dynamics of the decision process change if the stimulus values are changed, even if just their absolute values are manipulated while their difference is held constant [e.g., 21,22], each additional stimulus condition would have effectively halved the number of useable trials. However, one concern which stems from this decision is that participants may have adopted a 'detection' rather than 'discrimination' strategy. That is, participants may have initially focused on just one stimulus and made an initial 'detection' decision as to whether this was the brighter/darker stimulus. Then subsequently, they may have evaluated this via a more comparative judgement process. However, two factors decrease the likelihood of this possibility: firstly, the high degree of variability within the stimuli, relative to their mean difference, and secondly, the fact that both stimulus distributions were truncated at 1 SD from the mean to avoid extremely bright or dark stimulus values. Moreover, since participants' initial response times were well fit by a model which assumed a bounded evidence accumulation process, a hybrid-decision process does not offer the most parsimonious account of our findings. Nevertheless, future research is ultimately needed to fully examine the possibility of participants adopting a hybrid decision process in certain tasks or contexts.

We also chose to record responses in a binary rather than continuous fashion. This allowed us to precisely measure the onset times of change-of-mind responses, adding further constraint on the computational models. However, continuous response measures, such as movement trajectories, have been employed in a number of past studies [3,5,7,8] and have the advantage of allowing for changes of mind to be observed in a potentially more graded fashion. Moreover, physical effort costs are likely to differ for discrete and continuous responses, influencing the computations underlying change-of-mind decisions [7,8]. One question which arises in light of these considerations is whether the use of discrete response measures may have led to slightly different decision-making behaviour. For example, when continuous response measures are employed, participants may adopt different strategies because movements can be initiated before a strict decision has been finalized, and because they may be weighing up effort costs which evolve over time [7,8,23]. However, the general pattern of changes of mind we observed in this study, and in another recent study, which also employed a binary response measure [22], mirrors those observed in studies employing continuous response measures [3,5,7,8]. In particular, across all these studies changes of mind were more common following errors and were driven by stereotyped shifts in stimulus evidence. These commonalities suggest a common underlying process, in line with the dominant view that a

general process explains changes of mind across perceptual decision tasks [3]. Ultimately however, further research is needed to properly uncover the degree to which decision-making processes differ across binary and continuous responses.

To conclude, we have shown that pre-decisional evidence does influence the likelihood and speed of perceptual change-of-mind decisions. In particular, we have shown that the strength of even the very first frame of evidence one receives is associated with the speed and likelihood of later decision reversals. Moreover, we have shown that this finding can be accounted for by an extended diffusion decision model, in which initial sensory evidence exerts a slowly decaying bias on the decision process. This suggests that an initial 'snapshot' of sensory evidence biases subsequent sensory evidence accumulation, thus influencing later self-corrective behaviour.

## Materials and method

### Ethics statement

The experimental procedure was approved by the University of Melbourne ethics committee (ID 1749951.1). All participants gave informed written consent prior to the beginning of the experiment.

### Participants

Five people gave written informed consent and participated in the experiment. Participants had normal or corrected to normal vision and were aged 22–35 years (M = 26.4, SD = 5.32, 2 Female). Two participants were authors on this study (WT and DF). The others had pre-existing relationships with the authors, but were naïve as to the purpose of the experiment. Each participant completed 5 sessions of the experiment (except WT who complete 4.5 sessions due to a technical fault). Each participant was remunerated $15 AUD per session (except WT who did not receive payment). One participant was excluded from the final sample as they failed to respond in 19.44% of trials (the remaining participants only failed to respond in 1.9–3.1% of trials). The final sample was aged between 22–28 years (M = 24.25).

### Materials

All stimuli were presented on a Sony Trinitron Multiscan G420 CRT Monitor (Resolution 1280 x 1024 pixels; Frame Rate 75 Hz). Responses were recorded using a Tesoro Tizona Numpad (Polling Rate 1000 Hz). The task was coded in MATLAB 2015b using functions from the Psychophysics Toolbox Version 3.0.14 [24,25]. Whilst performing the experiment participants were seated in a dark room with their chin on a chinrest ~65cm from the screen.

### Procedure

In each experimental session, participants performed 1000 trials of a luminance discrimination task (Fig 4). On each trial of the task they indicated which of two flickering greyscale squares (70 x 70 pixels; ~2.18 x 2.18 degrees of visual angle) was the brightest. The squares were presented side-by-side with 70 pixels separating them horizontally. Participants made their responses with their left and right index fingers on the 1 (left response) and 3 (right response) keys of the numpad. Participants had 800 ms from stimulus onset to make an initial response. From the time of the initial response, the stimuli remained on screen for a fixed duration of 1.5 s. During this time participants were free to change their response. Participants were told to try to be as accurate as possible in their initial responses and to correct any errors they felt they had made. At the end of each trial, feedback, corresponding to the final response

**Fig 4. Schematic of the trial structure in the luminance judgement task.** Each trial began with the presentation of a red fixation dot for 500 ms. The flickering stimuli were then presented, and participants were given up to 800 ms to respond. From the time of the initial response the post-decision period began, lasting for a fixed duration of 1.5 s. Feedback was then presented for 300 ms in the form of ('correct' or 'error'). If participants failed to respond within 800 ms of the stimuli being presented, the post-decision period was skipped and 'too slow' was presented for 300 ms.

participants had made ("correct", "error" or "too slow"), was presented for 300 ms. A red fixation dot was presented for 500 ms before stimulus presentation. Self-paced breaks were provided every 100 trials.

The mean RGB values for the brighter and dark squares were 142 and 130 respectively. On each frame, independent greyscale values for the two stimuli were drawn from separate Gaussian distributions centered around their respective mean values. The standard deviation of the distributions was 55 and the distributions were truncated at 1 standard deviation.

## Psychophysical reverse correlation analysis

With each screen refresh (i.e. every 13.33 ms) a random luminance value was added to the mean luminance of each square. This enabled the use psychophysical reverse correlation [11,12] to reveal participants 'psychophysical kernels' (their information usage patterns) across time. The logic behind this analysis is as follows: if the residual luminance fluctuations at each frame are averaged across all trials they will cancel to zero, because they are randomly distributed. However, if the fluctuations systematically affect participants' decisions, then averaging across trials with shared decision outcomes will reveal how, and when, the fluctuations influenced participants' decisions.

To calculate participants' psychophysical kernels, the frame-by-frame luminance values for the darker stimulus were subtracted from those of the brighter stimulus. The across-trial mean difference (i.e. the difference in mean luminance between the two squares) was then subtracted, and the residual luminance fluctuations were normalized between -1 and 1. These fluctuations were sorted into trials which shared a response outcome (e.g., purely correct) or response characteristic (e.g., fast corrected errors). When sorting the trials by change of mind speed we calculated median change-of-mind response times (relative to the time of the initial response) for each participant, within each testing session. After sorting by trial type, the fluctuations were pooled across participants and averaged (Fig 1). To obtain the results in S2 Text, fluctuations for each trial type of interest were averaged within each participant. For the response-locked and change-of-mind-locked kernels, time points where there were fewer than 100 trials were excluded, to avoid noisy estimates.

## Statistical analyses

Trials in which participants failed to respond (~2.6% of trials on average per session), or in which they changed their mind more than once (~3.2% of trials) were excluded from all analyses. Trials in which the change-of-mind response time was less than 50 ms were also excluded (~0.0004% of trials). We also screened for trials in which the initial response time was less than

150 ms (no trials were rejected). Linear mixed effects models were used to analyse the data via the *lme4* [version 1.1, 26] package in R (version 3.5). A generalized linear mixed effects model was used to predict changes of mind, with main effects for initial decision accuracy and the first frame sensory evidence, as well as an interaction between initial decision accuracy and the first frame of sensory evidence. A linear mixed effects model was used to predict the time at which a change of mind occurred relative to the initial response (i.e. change of mind speed) with main effects for initial decision accuracy and the first frame sensory evidence, as well as an interaction between initial decision accuracy and the first frame of sensory evidence. Likelihood ratio tests were used to formally examine the effects of interest (i.e. the interaction between initial evidence and initial accuracy). The distribution of response times for changes of mind was more normally distributed that typical initial RT distributions. We therefore analysed these responses with a linear mixed effects model. In all models, a random intercept for participant was included. Code to reproduce all of these analyses is available at https://osf.io/a6u4n/.

## Computational modeling

We fit a variant of the extended DDM [3] to the response proportions and response time quantiles (0.1 0.3 0.5 0.7 0.9) of both the initial responses and the change of mind responses simultaneously. We simulated a discrete approximation of the extended DMM model (500,000 trials per iteration) and used the fminsearch algorithm (MATLAB 2016a) to minimize the root mean squared error between the simulated data and the actual data. Code for this model is available at https://osf.io/a6u4n/.

In our variant of the extended DDM, sensory evidence is noisily accumulated between two decision boundaries (0 and *B*). The average starting point of the accumulation process is halfway between the decision thresholds (i.e. *B*/2). From trial-to-trial the starting point varies uniformly around this average with a range of *Sz*. Once an initial decision threshold is reached, the evidence accumulation process continues to unfold. Note, that as in the original version of the extended DDM, there was a time limit parameter *timeOut* which specified the proportion of the post-decisional period for which participants processed additional information. If, during this period, enough evidence is accumulated against the initial decision such that a change-of-mind threshold–at distance of $B_{CoM}$ away from the initial decision threshold–is crossed, then a change of mind occurs.

With each timestep, the decision variable is updated as follows:

$$\Delta DV = drift * stepsize + noise * \sqrt{stepsize}$$

where, *drift* denotes the drift rate at a given time point, *noise* denotes the within trial noise at a given time point, and *stepsize* denotes the magnitude of the simulated timesteps within the model (0.001 s).

The drift rate at a given timepoint was determined as follows:

$$drift = mu + externalVar(t) + internalVar$$

where, *mu* denotes the mean drift rate, *externalVar* is the externally driven drift rate variability component at a given time point (this varies across time, see below), and *internalVar* is the internally driven across-trial drift rate variability component (which is constant across time). The externally driven across-trial drift rate variability component was determined as follows:

$$externalVar = -slope * t + s * firstFrame$$

Where, *slope* specifies the slope of a linearly decreasing function across *t* (time within a trial), *s* is a scaling parameter which is used to weight the internal representation of the first frame of sensory evidence (*firstFrame;* see below for details as to how sensory evidence is specified in this model). The internally driven across-trial drift rate variability parameter (*internalVar*) is a normally distributed random variable with a mean of zero and a standard deviation *eta*.

At each time-step the trial-specific drift rate is affected by within-trial noise. This can be thought of as a noisy representation of the stimulus flicker, which is determined as follows:

$$noise = stimulusNoise + N(0, 0.1)$$

where, *stimulusNoise* is a normally distributed random variable with a mean of zero and a standard deviation of *theta*. Like the luminance fluctuations in the real experiment, the stimulus noise in the model was truncated at 1 standard deviation. Conceptually, the *stimulusNoise* parameter models the frame-by-frame fluctuations in stimulus evidence which influence participants' decisions (see Fig 1).

Endogenous within-trial noise was modelled as a normally distributed random variable with a mean of zero and a standard deviation of 0.1. The standard deviation was fixed to 0.1 to act as a scaling parameter. Conceptually, this noise term accounts for within-trial sources of noise which were not stimulus driven (e.g., variability in moment to moment neural firing).

When determining the response times for initial responses and change-of-mind responses, we made the simplifying assumption that the non-decision time *tnd* and non-decision time variability *tndVar* components were the same for the two types of responses.

## Model-based psychophysical reverse correlation analysis

To understand the effects of pre-decisional evidence on participants' behaviour, it was important to derive model-predicted kernels, which could be compared with the participants' actual psychophysical kernels. By comparing these kernels, it was possible to test whether our new model variant was using 'sensory' evidence in the same way as the participants.

To construct model-predicted psychophysical kernels, we simulated 100,000 experimental trials from each model of interest. Critically, each model contained simulated stimulus noise (see section on *stimulusNoise* above), representing the moment-to-moment fluctuations in stimulus evidence that participants saw. After simulating, we sorted the trials into the four possible response outcome types, based on the model-predicted response. We then took the simulated stimulus noise for each trial and time-locked this to stimulus onset, the initial response, and the change of mind. Finally, we then averaged this noise across trials with shared decision outcomes, yielding the model-based psychophysical kernel estimates.

## Supporting information

**S1 Text. Marginal effects plots and parameter estimates from the mixed-effects models.**
(PDF)

**S2 Text. Individual-level figures and analyses.**
(PDF)

**S3 Text. Model predictions and parameters.**
(PDF)

**S4 Text. Auxiliary analyses of the overall effect of pre-response evidence.**
(PDF)

**S5 Text. Auxiliary analysis of trials in which the signal favoured the incorrect response** (PDF)

**S6 Text. Parameter estimates for 1–200 ms and 200–400 ms mean evidence models.** (PDF)

## Acknowledgments

We thank Milan Andrejević for helpful discussions.

## Author Contributions

**Conceptualization:** William Turner, Daniel Feuerriegel, Stefan Bode.

**Data curation:** William Turner.

**Formal analysis:** William Turner.

**Funding acquisition:** Robert Hester, Stefan Bode.

**Investigation:** William Turner.

**Methodology:** William Turner, Daniel Feuerriegel.

**Project administration:** William Turner, Daniel Feuerriegel, Stefan Bode.

**Software:** William Turner.

**Supervision:** Daniel Feuerriegel, Robert Hester, Stefan Bode.

**Writing – original draft:** William Turner.

**Writing – review & editing:** Daniel Feuerriegel, Robert Hester, Stefan Bode.

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
