## [Decision Letter · Decision Letter 0]

8 May 2021

Dear Mr. Turner,

Thank you very much for submitting your manuscript "An initial ‘snapshot’ of sensory information biases the likelihood and speed of subsequent changes of mind" for consideration at PLOS Computational Biology.

As with all papers reviewed by the journal, your manuscript was reviewed by members of the editorial board and by two independent reviewers. We had a hard time finding reviewers and apologize for the delay in making the initial decision. Both reviewers think this is an interesting study, which might constitute a novel contribution to the field. However, reviewer 1 has some concerns that need to be address. In light of the reviews (below this email), we would like to invite the resubmission of a significantly-revised version that takes into account the reviewers' comments.

We cannot make any decision about publication until we have seen the revised manuscript and your response to the reviewers' comments. Your revised manuscript is also likely to be sent to reviewers for further evaluation.

Sincerely,

Woo-Young Ahn

Associate Editor

PLOS Computational Biology

Samuel Gershman

Deputy Editor

PLOS Computational Biology

Reviewer's Responses to Questions

**Comments to the Authors:**

Reviewer #1: In this paper, the authors explore how the quantity of evidence accumulated before making a choice continues to impact the likelihood of changing one’s mind later on. Participants were presented two stimuli fluctuating in luminosity around two sets mean values. On each trial, they had to decide which of the two stimuli was the brightest. After the initial choice, the stimuli continued to be displayed, leaving the possibility to participants to change their mind. Reverse correlating the stimuli fluctuations to the choice revealed that early evidence had a long lasting impact on choice, as well as on the likelihood and speed of changes of minds. The authors used an extension of the Drift Diffusion Model to explain how early evidence can be used as a snapshot to which further evidence is compared and used to decide to change one’s mind.

The study is well conducted and certainly provides interesting new evidence on reversals of decisions. The paper is well written and easy to read, despite the complexity of the modelling and the statistics used. The main effect reported in the paper, namely the extension of primacy bias beyond choice to the timing and likelihood of changes of minds, is novel and certainly would constitute a significant contribution to the field. I also found the modelling approach appropriate and interesting, allowing to capture the behavioural data in a comprehensive manner.

My main concern however is the generalizability of the main finding of a long lasting effect of an early snapshot of evidence on changes of mind. Although primacy effects have been reported in the literature, such effect has not been observed on previous studies investigating changes of minds. Considering the novelty of the effect, it seems necessary to test further its robustness and explore alternative hypothesis. In particular, although reverse correlation is a powerful tool to understand the dynamics of decisions process, it is also notoriously hard to interpret and can lead to biases in interpretation. As the whole paper is centred on reverse correlation to proove the presence of a primacy effect on changes of mind, I find that additional analysis and/or experiments would be necessary to justify its existence. I list below additional comments and concerns.

- Test of the primacy effect: As I explained above, I believe further tests should be performed to understand the relative effect of the initial snapshot of evidence compared to future evidence on changes of mind. In particular, I think different GLMs including alternative predictors such as the overall pre-decisional signal strength or the evidence immediately preceding the response should be tested to establish the best model predicting the occurrence of changes of minds.

- Generalizability of the primacy effect to different paradigms: One important question raised by these findings is how likely such primacy effect is going to be observed for other types of decisions and stimuli. The authors do not discuss whether such effect is specific to the luminance task they used. Indeed, luminance could be seen as a stimulus feature that by definition requires less temporal integration than motion perception for instance, explaining why early momentary evidence might have such strong influence on choice and changes of mind. I would suggest to test whether such pattern of results would be observed in RDK stimuli or other stimulus discrimination task. This is important to understand the scope of the findings. A paragraph on this should also be added in the discussion.

- Another concern is the fact that the authors used constant means for the low and high luminance stimuli. Therefore, it is possible for the participants to perform the task not as a discrimination task but as a detection task, learning to detect the presence or absence of the high luminance stimulus on one side of the screen only. Indeed, the presence of the primacy effect could be explained by an early “detection strategy” where the presence of high luminance signal at the onset of the trial in one of the stimuli would lead participants to believe they have detected the presence of the high luminance stimulus, while later accumulation of evidence would switch to a comparison of the evidence of the 2 stimuli. To avoid participants using such strategy, it would have been interesting to vary in the design the overall mean luminance of the two stimuli, while keeping the difference in luminance between them constant. That would have allowed to determine whether the primacy effect on changes of mind is only a by-product of the task design and participants optimizing their choice process for that design. I would suggest such experiment is added to the design and this hypothesis is discussed in the paper too.

- One important potential limit of reverse correlation analysis is to neglect the effect of signal mean on choice, as it focusses only on fluctuations above and below the stimulus mean. However, the mean difference between the two signals has certainly an effect on choice. I would suggest that the authors provide supplementary figures showing the unaltered values of signal fluctuations in each of the luminance stimuli in a supplementary figure. That would allow to understand better what is the actual signal feeding into the decision and changes of mind. In particular, this would allow to determine whether the two stimuli have a symmetrical effect on choice and changes of minds.

- Using the same line of reasoning, I would suggest that the authors provide further analysis to determine the proportion of trials where, by chance, the overall signal favoured the incorrect response. Is it possible that they drive the reverse correlation effect and some very types of changes of mind? These trials may be difficult to analyse and classify as correct or incorrect, and should probably be removed from further analysis or analysed separately.

- Another question that arises from the present finding and modelling developed is what exactly the primacy effect reported means in terms of weight given to each sample of information. In particular, while the reverse correlation can be interpreted as early evidence having stronger effect on changes of mind and therefore being weighted more in the evidence accumulation process, the converse interpretation is also plausible. Namely, the signal fluctuations in the early presentation of the stimulus are weighted less and therefore need to be stronger to have an impact on changes of mind, which is why they appears as deviating from the mean in the reverse correlation time-series. Can the authors comment/discuss this?

- Reverse correlation methods: It is unclear from the methods how the analysis dealt with the absence of data points in the reverse correlation time-series. Presumable in the stimulus locked graph later time-points are less populated? Similarly, in response-locked time-series, early time points are less populated. The authors should report how they dealt with the change in number of trials forming the reverse correlation time-courses and how it could have affected their results.

- In Figure 3D&H, the authors report no differences in the psychophysical kernels for the first frame of evidence in the main text. However, a clear effect is observed on the figure. Can the authors explain?

Reviewer #2: In this experiment, the authors examine changes of mind in a simple perceptual decision.  They find that the early evidence (even the very first frame of evidence) has a subsequent influence on both the likelihood of a subsequent change-of-mind, and the speed at which that change-of-mind occurs. The authors propose a variant on the extended diffusion decision model in which initial sensory information has a slowly decaying bias on subsequent evidence accumulation.

This is a very interesting paper on an important topic. The novel approach to investigating the influence of initial information and data from this approach are a valuable contribution to the field. This also provides some important updates to an influential model of perceptual decision-making and changes-of-mind. A few comments:

- One key difference that might warrant at least a bit of discussion - the current approach uses a discrete binary response (a keypress) whereas a number of the studies cited in the introduction use a continuous response such as cursor movement, joystick movement, hand movement, etc.  In those studies, changes-of-mind were often defined as a change in movement path. This may not be a major issue in interpreting the present data but it may be worth addressing somewhere in the manuscript - it is possible that decision-making processes would differ when continuous movements (which more easily allow for the initiation of movement before an initial decision is really finalized; e.g., Gallivan & Chapman, 2014) are involved.

- It might be helpful to have a bit more discussion about the other data (e.g., the change-of-mind locked data) aside from the first frame of evidence, in addition to the figures that are presented. For example, is there anything worth discussing in Fig 1C and 1F?

- t-tests are conducted in Figure 1, but more detail could be provided as to how those are calculated. Are they based on participant mean scores for residual evidence at each point?

- line 194-195 has a typo: "As such, change-of-mind decisions depend entirely on the quality of post-decisional.

- typo, line 383: "were excluded from all analyse."

**Have the authors made all data and (if applicable) computational code underlying the findings in their manuscript fully available?**

Reviewer #1: Yes

Reviewer #2: Yes

PLOS authors have the option to publish the peer review history of their article (what does this mean?). If published, this will include your full peer review and any attached files.

Reviewer #1: **Yes: **Lucie Charles

Reviewer #2: No
---

## [Decision Letter · Decision Letter 1]

17 Sep 2021

Dear Mr. Turner,

Thank you very much for submitting your manuscript "An initial ‘snapshot’ of sensory information biases the likelihood and speed of subsequent changes of mind" for consideration at PLOS Computational Biology. As with all papers reviewed by the journal, your manuscript was reviewed by members of the editorial board and by several independent reviewers. The reviewers appreciated the attention to an important topic and think the authors have addressed their comments on an earlier version. Based on the reviews, we are likely to accept this manuscript for publication, providing that you modify the manuscript according to the review recommendations. Please check the comments by the two reviewers, especially Reviewer #1 who requested an additional analysis (R1C2). 

Sincerely,

Woo-Young Ahn

Associate Editor

PLOS Computational Biology

Samuel Gershman

Deputy Editor

PLOS Computational Biology

[LINK]

Reviewer's Responses to Questions

**Comments to the Authors:**

Reviewer #1: The authors have made a very good effort to address my initial sets of comments and most of the issues I raised are now resolved. I still believe there are some points regarding the primacy effect that need to be addressed however.

I appreciate the authors are not claiming that only the first-frame of evidence influenced the occurrence of changes of mind (fortunately). However, as it seems it is the main point that they are focussing on and which justifies the novelty of the paper (as clearly emphasized in the title and abstract), I still find that some analysis are missing to understand the relevance of this effect for changes of mind. I agree with the authors it is novel and worth reporting, however I think it is equally important not to be misleading in the interpretation of the results and to report clearly the relative importance of this primacy effect compared in the evidence accumulation process leading to changes of mind.

I detail below how I think the authors should address this.

R1C2

I appreciate the authors have run additional analysis confirming that the mean pre-response evidence predicted the probability and latency of changes of minds.

1/ I think this is an important result that should be moved from the S4 appendix to the main text.

2/ I think an additional analysis should be run to compare the predictive power of the initial frame of evidence and the rest of the pre-response window. According to figure S7, it looks like the mean of the pre-response window is a better predictor of occurrence of changes of minds than the initial snapshot singled out by the authors. However, this should be tested and quantified. As the mean of the whole pre-response window is not comparable to evidence presented in only one frame, I would suggest using a different approach than computing the mean pre-response evidence.

One possibility would be to run the same GLM analysis comparing the predictive power of the initial snapshot, and then another sample 100 or 200ms later, correcting for multiple comparison. Another possibility would be to compute the mean over the 0-200ms time window and compare it to the 200-400ms for instance.

R1C3

I thank the authors for their detailed reply and additional results presented. One thing that I think might be worth mentioning in the discussion about the generalisability of the effect is the issue of temporal expectancy. As in the design, the stimulus appeared always at predictable time following the start of the trial, could it be that participants learn to focus their attention to when the initial frame of the stimulus would appear?

R1C6

Thanks for analysing the proportion of trials in which the signal favoured the incorrect response. I agree the proportion is small enough to have a negligible effect but it is worth reporting it still in the supplementary material.

R1C7

Thanks for reporting this interesting finding of the model with the down weighting of the initial evidence. I think that strengthen the point made in the paper and is well highlighted in the updated part of the discussion.

Reviewer #2: The authors have done an excellent job addressing the comments made on the initial submission, and i think this paper is a strong contribution to the field.  I only have a couple minor comments:

1. The figures included in this version are quite blurry - hopefully this is something the authors can fix in the final submission.

2. line 285 typo: "relative rarity of these trial" should be "trials"

3. for the added paragraph about continuous responses vs. discrete responses in the discussion, it might be worth also noting that continuous responses involve more physical effort which may be a contributing factor to the likelihood and nature of changes of mind (several studies showing this are already cited in this paragraph).

4. Regarding the discussion on temporal integration and the RDK tasks, it may be worth discussing or speculating on real-world examples where changes-of-mind may occur that do (or do not) involve temporal integration. In other words, regarding R1's concerns about generalizability, perhaps the initial snapshot of evidence is only useful in a subset of cases. Is it likely that many real-world behaviors involving changes of mind are covered by these cases?

**Have the authors made all data and (if applicable) computational code underlying the findings in their manuscript fully available?**

Reviewer #1: Yes

Reviewer #2: Yes

PLOS authors have the option to publish the peer review history of their article (what does this mean?). If published, this will include your full peer review and any attached files.

Reviewer #1: No

Reviewer #2: No

Figure Files:

Data Requirements:

Reproducibility:

References:

---

## [Decision Letter · Decision Letter 2]

9 Dec 2021

Dear Mr. Turner,

We are pleased to inform you that your manuscript 'An initial ‘snapshot’ of sensory information biases the likelihood and speed of subsequent changes of mind' has been provisionally accepted for publication in PLOS Computational Biology.

Best regards,

Woo-Young Ahn

Associate Editor

PLOS Computational Biology

Samuel Gershman

Deputy Editor

PLOS Computational Biology

Reviewer's Responses to Questions

**Comments to the Authors:**

Reviewer #1: I thank the authors for their careful consideration of my comments and their detailed reply.

I think it is fine to leave Figure S7 is the supplementary material considering the result of the corresponding analysis is now mentioned in the main text.

I appreciate that the authors have now performed the analysis of the rest of the pre-response time-window and thank them for their interesting comment regarding their interpretation of the finding.

One could consider that the result that the 200-400ms time window is a better predictor of changes of mind than the early snapshot of evidence does weaken the main point of the paper. However, I think that the fact that this is now explicitly stated and discussed in the main text means an attentive reader cannot be misled into misinterpreting the effect of early evidence on changes of mind. And I fully agree with the authors that the fact that the first frame of evidence is predictive of later changes of mind is a theoretically interesting finding and worth reporting.

I leave the editor to review this last point but it seems to me that the paper makes an interesting and thought-provoking contribution to the field and I therefore recommend publication.

**Have the authors made all data and (if applicable) computational code underlying the findings in their manuscript fully available?**

Reviewer #1: Yes

PLOS authors have the option to publish the peer review history of their article (what does this mean?). If published, this will include your full peer review and any attached files.

Reviewer #1: **Yes: **Lucie Charles

---

## [Editor Report · Acceptance letter]

17 Dec 2021

PCOMPBIOL-D-21-00441R2 

An initial ‘snapshot’ of sensory information biases the likelihood and speed of subsequent changes of mind

Dear Dr Turner,

I am pleased to inform you that your manuscript has been formally accepted for publication in PLOS Computational Biology. Your manuscript is now with our production department and you will be notified of the publication date in due course.

With kind regards,

Anita Estes
